# Exploring Human-in-the-Loop Test-Time Adaptation by Synergizing Active Learning and Model Selection

**Yushu Li**[*][1,2]   **Yongyi Su**[*][1,2]   **Xulei Yang**[2]   **Kui Jia**[3]   **Xun Xu**[†][2]

[1]**South China University of Technology, Guangzhou, China**
[2]**Institute for Infocomm Research ($I^2R$), A\*STAR, Singapore**
[3]**School of Data Science, The Chinese University of Hong Kong, Shenzhen, China**

*Reviewed on OpenReview:* `https://openreview.net/forum?id=PO9rAv8UH7`

## Abstract

Existing test-time adaptation (TTA) approaches often adapt models with the unlabeled testing data stream. A recent attempt relaxed the assumption by introducing limited human annotation, referred to as Human-In-the-Loop Test-Time Adaptation (HILTTA) in this study. The focus of existing HILTTA studies lies in selecting the most informative samples to label, a.k.a. active learning. In this work, we are motivated by a pitfall of TTA, i.e. sensitivity to hyper-parameters, and propose to approach HILTTA by synergizing active learning and model selection. Specifically, we first select samples for human annotation (active learning) and then use the labeled data to select optimal hyper-parameters (model selection). To prevent the model selection process from overfitting to local distributions, multiple regularization techniques are employed to complement the validation objective. A sample selection strategy is further tailored by considering the balance between active learning and model selection purposes. We demonstrate on 5 TTA datasets that the proposed HILTTA approach is compatible with off-the-shelf TTA methods and such combinations substantially outperform the state-of-the-art HILTTA methods. Importantly, our proposed method can always prevent choosing the worst hyper-parameters on all off-the-shelf TTA methods. The source code is available at `https://github.com/Yushu-Li/HILTTA`.

## 1 Introduction

Deep learning has demonstrated remarkable success in numerous application scenarios. Nevertheless, the disparity in distribution between training and testing data leads to poor generalization of deep neural networks. In many real-world scenarios, the data distribution of testing data is often unknown until the inference stage (Sun et al., 2020; Wang et al., 2020) and the distribution may experience constant shift over the course of inference (Wang et al., 2022; Yuan et al., 2023; Press et al., 2023; Lee & Chang, 2024; Su et al., 2024a). These practical challenges render traditional unsupervised domain adaptation and source-free domain adaptation paradigms ineffective. To overcome the above challenges, test-time adaptation (TTA) emerges by adapting model weights upon observing testing data at inference stage (Wang et al., 2020; Sun et al., 2020; Wang et al., 2022; Su et al., 2022; 2024a). The success of TTA is often attributed to the practice of aligning testing data distribution with source domain distribution (Su et al., 2022; Kang et al., 2023; Wang et al., 2023a) and/or self-training on pseudo labels (Jang et al., 2023; Su et al., 2024b; Marsden et al., 2024a). Despite achieving impressive results on a wide range of tasks, e.g. classification (Sun et al., 2020), image segmentation (Wang et al., 2023b) and object detection (Chen et al., 2023), the fully unsupervised TTA paradigm is still overwhelmed by remaining challenges including hyper-parameter tuning (Zhao et al., 2023), continual distribution shift (Wang et al., 2022), non-i.i.d. (Gong et al., 2022; Niu et al., 2023; Yuan et al.,

---

[*]Equal contribution. This work was done during Yushu Li and Yongyi Su's attachment with $I^2R$ as visiting students.
[†]Corresponding author. Email: alex.xun.xu@gmail.com

2023), class imbalance (Su et al., 2024a), etc. Unfortunately, there are still no principled solutions to address these challenges without relaxing the assumptions.

Figure 1: An illustration of Human-in-the-Loop Test-Time Adaptation framework. Upon collecting a batch of testing data $\{x_i\}_{i=1\cdots N_b}$, we train multiple candidate models $\theta^*(\omega_m)$ using candidate hyper-parameters $\{\omega_m\}$. We further select candidate samples for annotation through K-Margin. Model selection is achieved by choosing the hyper-parameter minimizing validation objective. We repeat until the testing data stream concludes, following a continual TTA framework.

Recently, efforts have been made to the exploration of introducing a limited annotation budget during test-time adaptation (SimATTA) (Gui et al., 2024). In this study, we refer to this concept as human-in-the-loop test-time adaptation (HILTTA). Through a theoretical analysis, SimATTA (Gui et al., 2024) demonstrated that adapting pre-trained model with carefully selected labeled samples on the testing data stream could substantially improve the effectiveness of TTA and prevent catastrophic forgetting (Kemker et al., 2018; Li & Hoiem, 2017) upon adapting to continuously changing testing domains (Wang et al., 2022). Apart from proving the theoretical guarantees, the main focus of SimATTA (Gui et al., 2024) lies in developing a balanced incremental clustering approach to select the most informative samples on the testing data stream for annotation. Specifically, SimATTA takes into consideration the model uncertainty and feature diversity for selecting the most informative samples to annotate on the testing data stream.

Despite the theoretical guarantee and superior empirical observations, the underlying practice of utilizing label budget by existing active HILTTA methods follows the traditional line of research on active learning (Sener & Savarese, 2018; Saran et al., 2023). In this study, we aim to explore the question: **How can human-in-the-loop test-time adaptation be realized beyond an active learning perspective?** To answer this question, we first revisit the pitfalls of state-of-the-art test-time adaptation (Zhao et al., 2023). Among the multiple remaining pitfalls, we pinpoint hyper-parameter selection

Table 1: The differences between our proposed human-in-the-loop test-time adaptation (HILTTA) and existing TTA settings.

| Setting | Active Learning | Model Selection |
|---|---|---|
| Test-Time Adaptation (TTA) | - | - |
| Active Test Time Adaptation (ATTA) | ✓ | - |
| Human-in-the-Loop TTA (HILTTA) | ✓ | ✓ |

as the most nasty one due to the sensitivity to the choice of hyper-parameters by many TTA methods. Hyper-parameter selection, also referred to as model selection, aims to choose the optimal hyper-parameters for model training. Established paradigms towards model selection often rely on a held-out validation set and the model selection task is defined as discovering the hyper-parameters trained upon which the model

displays optimal generalization performance on the held-out validation set (Yang & Shami, 2020). Inspired by the success of hyper-parameter optimization, we offer a fresh perspective on human-in-the-loop test-time adaptation by synergizing active learning and model selection. In specific, we present an HILTTA procedure with three steps. First, upon observing a testing sample, the model will make an instant prediction to minimize inference latency. When a fixed number of testing samples are observed, we employ a data selection strategy to recommend a subset of testing samples for human oracle to label. The labeled data will then be utilized as the validation set for selecting the best hyper-parameters. Finally, the model is further updated with the labeled data in a fully supervised fashion. With the novel HILTTA steps, we intend to make better utilization of the limited annotation budget compared to existing HILTTA approaches. We demonstrate the differences between our proposed HILTTA and related TTA settings in Tab. 1.

To enable the novel HILTTA procedure, we identify two major challenges. First, the validation objective deserves delicate design due to the small validation set. In particular, without seeing the whole picture of testing data, a naive model selection strategy may overfit to the labeled testing data within a local time window, resulting in unstable and biased hyper-parameter selection. Second, the strategy for selecting testing samples to annotate must be well-calibrated for the task of HILTTA. Employing the data acquisition functions developed for generic active learning (Sener & Savarese, 2018) or active testing (Kossen et al., 2022) is suboptimal.

To tackle these challenges, we first propose a hyper-parameter regularization technique by considering the deviation of model predictions from a frozen source model. We further combine the ranking scores of validation loss and the deviation as the scoring function for model selection. To accommodate the active learning strategy for HILTTA from a model selection perspective, we prioritize both model uncertainty and sample diversity and combine the two objectives through a simple re-weighting. Sample selection is then achieved by maximizing the coverage in the re-weighted feature space. Extensive evaluations on TTA datasets demonstrate the effectiveness of the proposed HILTTA approach.

We summarize the contributions of this work as follows.

- To the best of our knowledge, we are the first to approach Human-In-the-Loop Test-Time Adaptation (HILTTA) by synergizing active learning and model selection and propose an off-the-shelf HILTTA method.

- To prevent the model selection from overfitting to a small validation set and local distribution, we introduce multiple regularization techniques. We further identify an effective selection strategy for HILTTA by considering the feature diversity and model uncertainty.

- We demonstrate that the proposed method is compatible with state-of-the-art TTA methods. The combined methods outperform dedicated active TTA methods and stream-based active learning methods on five test-time adaptation datasets.

## 2 Related Work

### 2.1 Test-Time Adaptation

The generalization of deep learning model is hindered by the distribution gap between source and target domain. Traditional approaches towards improving the generalization of deep learning models follow unsupervised domain adaptation (UDA) (Ganin & Lempitsky, 2015) which adapts model weights to learn indiscriminative features across source and target domains. Simultaneously training on both source and target domain data is infeasible in many real-world scenarios where access to source training data is prohibited due to privacy or storage overhead. Source-free domain adaptation (SFDA) targets this issue and adapts model weights to target domain data only (Liang et al., 2020; Liu et al., 2021; Yang et al., 2021; Liang et al., 2021). In contrast to the assumptions made in UDA and SFDA, test-time adaptation (TTA) emerges in response to the need for adapting pre-trained model to unknown target domain at inference stage (Su et al., 2022; Sun et al., 2020; Wang et al., 2020). TTA has been demonstrated to improve model's generalization to out-of-distribution testing data and the success owns to self-training on unlabeled testing

data. To prevent the model from being mislead by incorrect pseudo labels, a.k.a. confirmation bias (Arazo et al., 2020), people choose to update only a fraction of model weights, e.g. classifier weights (Iwasawa & Matsuo, 2021) and normalization layers (Wang et al., 2020), or introducing regularizations, e.g. distribution alignment (Liu et al., 2021; Su et al., 2024b) and representation regularization (Su et al., 2024a). More recent focus of test-time adaptation research lies on tackling challenges arising from real-world testing data, including continual distribution shift (Wang et al., 2022), class imbalance (Su et al., 2024a), update with small batchsize (Niu et al., 2023) and open-world testing data (Li et al., 2023; Zhou et al., 2023). Nevertheless, existing TTA approaches are mainly built upon a fully unsupervised adaptation protocol. We argue that sparse human annotation budgets might be available and exploiting annotation budget in TTA, referred to as human-in-the-loop TTA, has gained attention from the community (Gui et al., 2024). Contrary to the traditional active learning perspective (Gui et al., 2024) we identified an unexplored opportunity to treat labeled data as the validation set for model selection and we propose multiple techniques to improve the efficacy of human-in-the-loop test-time adaptation.

## 2.2 Active Learning

Active learning (AL) aims to select the most informative samples for Oracle to label for supervised training. Prevailing approaches often prioritize uncertainty (Gal & Kendall, 2017), feature diversity (Sener & Savarese, 2018) or the combination of both (Ash et al., 2020). The traditional AL approaches often adopt a batch-wise paradigm. When data arrive in a sequential manner stream-based AL (Saran et al., 2023) proposed to quantify the increment of the determinant as the criteria for incremental sample selection. Active learning was integrated into test-time adaptation to tackle the forgetting issue (Gui et al., 2024). In contrast to tackling HILTTA from a pure active learning perspective, we propose to synergize active learning and model selection. Selecting samples for model selection often relies on an orthogonal objective to active learning. In the related works, active testing (Kossen et al., 2021; 2022; Matsuura & Hara, 2023; Ochiai et al., 2023) was developed to select a subset of testing data that is representative of the whole testing data. Active testing has been demonstrated to help label-efficient model validation. Nevertheless, active learning and active testing will prioritize supervised training and model validation respectively. We believe either is not enough for the purpose of synergizing active learning and model selection for test-time adaptation.

# 3 Methodology

## 3.1 Human-in-the-Loop Test-Time Adaptation Protocol

We first provide an overview of the existing test-time adaptation (TTA) evaluation protocol. TTA aims to adapt a pre-trained model $h(x; \theta)$ to a target testing data stream $\mathbb{D} = \{x_i, \bar{y}_i\}_{i=1 \cdots N_t}$ where $x_i$ and $\bar{y}_i \in \{1 \cdots C\}$ denote the testing sample and the associated unknown label, respectively. Existing test-time adaptation approaches often make an instant prediction $\hat{y} = h(x; \theta)$ on each testing sample and then update model weights $\theta$ upon observing a cumulative batch of samples $\mathbb{B}_t = \{x_i\}_{i=1 \cdots N_b}$ where $t$ refers to the $t$-th minibatch. Prevailing methods enable adaptation by self-training (Wang et al., 2020; Su et al., 2024b) or distribution alignment (Su et al., 2022). Unlike the existing assumption of totally unsupervised adaptation, additional human annotation $\mathbb{B}_t^l = \{x_j, y_j\}_{j=1 \cdots N_b^l}$ within each batch could be introduced (Gui et al., 2024) and TTA is implemented on the combine of large unlabeled data $\mathbb{B}_t^u = \mathbb{B}_t \setminus \mathbb{B}_t^l$ and sparsely labeled data $\mathbb{B}_t^l$. We refer to the novel evaluation protocol as Human-in-the-Loop Test-Time Adaptation (HILTTA) throughout the study.

## 3.2 Model Selection with Sparse Annotation

We present the details for model selection with sparse human annotation on the testing data stream. The choice of hyper-parameters is known to have a significant impact on the performance of model training and generalization. W.l.o.g., we define the task of model selection by introducing a fixed candidate pool of hyper-parameters $\Omega = \{\omega_m\}_{m=1 \cdots N_m}$ with $N_m$ options. The choice of hyper-parameters hinges on the sensitivity of respective TTA methods, e.g. pseudo label threshold, learning rate, and loss coefficient are commonly chosen hyper-parameters for selection. We further denote the discriminative model trained upon

the hyper-parameters $\omega$ as $\theta(\omega)$. The model selection can be formulated as a bi-level optimization problem as follows, where $\mathcal{L}_{tr}(\cdot; \omega)$ denotes the training loss dictated by the hyper-parameter $\omega$, $\mathcal{L}_{val}(\cdot, \cdot)$ denotes the validation loss/objective and $\mathbb{D}^l$ refers to the labeled data.

$$\min_{\omega \in \Omega} \frac{1}{N_t^l} \sum_{(x_j, y_j) \in \mathbb{D}^l} \mathcal{L}_{val}(h(x_j; \theta^*(\omega)), y_j) \quad \text{s.t.} \quad \theta^*(\omega) = \arg\min_\theta \frac{1}{N_t} \sum_{x_i \in \mathbb{D}} \mathcal{L}_{tr}(h(x_i; \theta), \omega) \tag{1}$$

We interpret the above bi-level optimization problem as discovering the optimal hyper-parameter $\omega$ trained upon which the model weights $\theta^*$ achieves the best performance on the labeled dataset. Solving the above problem through gradient-based method (Liu et al., 2018) is infeasible for TTA tasks where adaptation speed is a major concern while gradient-based methods require iteratively updating between meta gradient descent $\nabla_\omega \mathcal{L}_{val}$ and task gradient descent $\nabla_\theta \mathcal{L}_{tr}$. Considering the task gradient descent only takes a few steps in a typical TTA setting, we opt for an exhaustive search for the hyper-parameters in the discretized hyper-parameter space. In specific, we enumerate the candidate models adapted with different hyper-parameters as $\{\theta_m^*\}_{m=1\cdots|\Omega|}$. The candidate with the best validation loss is chosen as the best model $\theta^*$, as follows.

$$\forall m = 1, \ldots, N_m, \quad \theta^* = \arg\min_{\theta \in \{\theta_m^*\}} \sum_{x_j, y_j \in \mathbb{D}^l} \mathcal{L}_{val}(h(x_j; \theta), y_j)$$
$$\text{s.t.} \quad \theta_m^* = \arg\min_\theta \frac{1}{N_t} \sum_{x_i \in \mathbb{D}} \mathcal{L}_{tr}(h(x_i; \theta), \omega_m) \tag{2}$$

The specific choice of training loss $\mathcal{L}_{tr}$ could be arbitrary off-the-shelf test-time adaptation methods while the design of validation loss $\mathcal{L}_{val}$ deserves careful attention in order to achieve robust model selection.

### 3.3 Design of Validation Objective

In this section, we take into consideration two principles for the construction of validation objectives. First, the validation loss should mimic the behavior of the model on the testing data stream. Assuming the limited labeled data takes a snapshot of the whole testing data stream and classification is the task to be addressed, an intuitive choice is the cross-entropy loss as follows.

$$\mathcal{H}_{ce}(x_i; \theta_m^*) = \sum_{c=1}^{C} \mathbb{1}(y_i = c) \log h_c(x_i; \theta_m^*) \tag{3}$$

Alternative to the continuous cross-entropy loss, accuracy also characterizes the performance on the labeled validation set. However, the discretized nature of accuracy renders this option less suitable due to many tied optimal hyper-parameters.

**Regularization for Stable Model Selection:** The stability of model selection heavily depends on the selection of the validation set and the variation of target domain distribution. Under a more realistic TTA scenario, the target domain distribution could shift over time, e.g. the continual test-time adaptation (Wang et al., 2022). Such a realistic challenge may render cross-entropy loss less effective due to over-adapting to local distribution. To stabilize the model selection procedure, we propose the following two measures.

**Anchor Deviation for Regularizing Model Selection:** We first introduce an anchor deviation to regularize the model selection procedure. This regularization is similar to the anchor network proposed in Su et al. (2024a) except that we use the anchor deviation to regularize model selection which is orthogonal to the purpose in Su et al. (2024a). Specifically, we keep a frozen source domain model, denoted as $\theta_0$. The anchor loss is defined as the L2 distance between the posteriors of the frozen source model and the $m$-th candidate model $\theta_m^*$. A big deviation from the source model prediction suggests the model has been adapted to a very specific target domain and further adapting towards one direction would increase the risk of failing to adapt to the continually changing distribution.

$$\mathcal{R}_{anc}(x_i; \theta_m^*) = \frac{1}{N_b} \sum_{(x_i,y_i)\in\mathbb{D}^l} \|h(x_i; \theta_0) - h(x_i; \theta_m^*)\| \tag{4}$$

**Smoothed Scoring for Model Selection:** To determine the optimal model to select, we combine the cross-entropy loss and the anchor deviation as the final score. Direct summing the two metrics as the model selection score is sub-optimal due to the relative scale between the two metrics. Therefore, we propose to first normalize the two metrics as $\text{Norm}(x) = \frac{x-\min(x)}{\max(x)-\min(x)}$. After the normalization, the relative size relationships of different candidate model scores are preserved, and a comparable validation loss can be conveniently obtained by addition, as below.

$$\mathcal{L}_{val}(x_i; \theta_m^*) = S_{ce}(x_i; \theta_m^*) + S_{anc}(x_i; \theta_m^*)$$
$$\text{s.t.} \quad \forall m = 1, \dots, |\Omega|, \quad S_{ce}(x_i; \theta_m^*) = \text{Norm}(\mathcal{H}_{ce}(x_i; \theta_m^*)), \quad S_{anc}(x_i; \theta_m^*) = \text{Norm}(\mathcal{R}_{anc}(x_i; \theta_m^*)), \tag{5}$$

Considering that the testing data stream is often highly correlated, i.e. temporally adjacent samples are likely subject to the same distribution shift, the optimal hyper-parameter/model should change smoothly as well. To encourage a smooth transition of the optimal model, we select the optimal model following an exponential moving average fashion as in Eq. 6 where we denote the moving average validation loss at the $t$-th batch as $\bar{\mathcal{L}}_{val}^t$ and the chosen best model $\theta^*$ is obtained as the one that minimizes the smoothed validation loss.

$$\bar{\mathcal{L}}_{val}^t(\theta_m^*) = \beta \bar{\mathcal{L}}_{val}^{t-1}(\theta_m^*) + (1-\beta)\frac{1}{|\mathbb{B}_t^l|} \sum_{x_j,y_j\in\mathbb{B}_t^l} \mathcal{L}_{val}(h(x_j; \theta_m^*), y_j),$$
$$\theta^* = \arg\min_m \bar{\mathcal{L}}_{val}^t(\theta_m^*) \tag{6}$$

### 3.4 Sample Selection for HILTTA

The sample selection strategy should balance the objectives of active learning and model selection for more effective HILTTA. We achieve this target from two perspectives. First, the selected samples should prioritize low-confidence samples for more effective active learning, as adopted by SimATTA (Gui et al., 2024) through incremental K-means. However, selecting redundant low-confidence samples may harm the effectiveness of model validation. This motivates us to select samples approximating the testing data distribution, similar to the objective of active testing (Kossen et al., 2021). Biasing towards either criterion would result in a suboptimal HILTTA. Our design mainly aims to balance the preference over selecting low-confidence and diverse samples.

A similar objective as above was achieved by a recent active learning approach, BADGE (Ash et al., 2020), which takes the gradient of loss w.r.t. the penultimate layer feature as the uncertainty weighted feature as $g_x = \frac{\partial}{\partial\theta_{out}}\mathcal{L}_{ce}(h(x; \theta), \hat{y})$, where $\theta_{out} \in \mathbb{R}^{D\times C}$ is the classifier weights. Hence, the gradient embedding $g_x \in \mathbb{R}^{D\cdot C}$ is of dimension $D \cdot C$. When the semantic space is substantially high, e.g. $C = 1000$ for ImageNet (Deng et al., 2009), the above gradient embedding results in extremely high dimension features. For instance, a typical penultimate layer feature is 2048 channels and the gradient embedding could reach 2,048,000 channels. Such a high dimension prevents efficient clustering operations, e.g. due to memory constraints, for sampling. To mitigate the ultra-high dimension challenge, we integrate uncertainty into feature representation by multiplying the uncertainty, measured by the margin confidence, to the feature representation as in Eq. 7, where $\text{sort}^d$ refers to the descending sorting.

$$\hat{p} = \text{sort}^d [h_1(x_i; \theta), h_2(x_i; \theta), \cdots, h_C(x_i; \theta)],$$
$$g_i = (1 - \hat{p}_1 + \hat{p}_2)f(x_i; \theta) \tag{7}$$

The above design prioritizes the difference between the probability of the most confident and second most confident classes as a quantification of model uncertainty. Such a design is superior to alternative designs, e.g. entropy, for the following reasons. First, deep learning models tend to be over confident with low entropy in general. Thus the gap between top 1 and top 2 class probabilities may better suggest the model confidence than entropy. Moreover, the entropy may become less distinct for a classification task with many categories as the randomness on non-ground-truth classes may affect the overall entropy calculation. Finally, our empirical observations in Sect. 4.3 also suggest the superiority of using the probability margin.

**Sample Selection via K-Margin**: The samples to be selected for annotation are eventually obtained by conducting a K-Center clustering on the uncertainty-weighted features within each minibatch, as in Eq. 8, which is referred to as K-Margin clustering. We select K samples $\mathbb{B}_t^l$ such that the maximal distance of all testing sample embedding is minimized. We deliberately exclude the selected samples in the previous minibatch, as opposed to SimATTA (Gui et al., 2024), because the labeled samples will serve as the validation set, and excluding new samples that may overlap with previously selected samples could undermine the effectiveness of model selection. The following problem can be solved via a greedy selection algorithm (Sener & Savarese, 2018).

$$\min_{\mathbb{B}_t^l \subseteq \{g_i\}_{i=1}^{N_b}} \max_{g_i \in \{g_i\}_{i=1}^{N_b}} \min_{c_k \in \mathbb{B}_t^l} ||g_i - c_k||, \quad s.t. \ |\mathbb{B}_t^l| \leq K \tag{8}$$

Our K-Margin approach with the K-Center clustering approach offers a computationally efficient alternative to K-Means++ while similarly balancing uncertainty and diversity in sample selection. Unlike K-Means++, which requires iterative center updates, K-Center clustering selects points that maximize coverage by minimizing the maximum distance from each sample to its nearest center. This method provides a representative subset with less computational overhead, which is more suitable in TTA settings. By focusing on high-uncertainty, diverse samples, K-Center clustering ensures a labeled subset that accurately reflects the test distribution. It guarantees that the labeled subset spans the feature space with fewer computations, thereby maintaining a consistent approximation of the test distribution. This enhances model selection and supervised training by creating a robust validation set that captures broad feature variance without redundant low-confidence samples.

### 3.5 Overall Algorithm for HILTTA

We propose two types of training losses for model updates and a validation loss for model selection. During the model selection stage, we generate multiple candidate models by training with an unsupervised TTA loss, denoted as $\mathcal{L}_{tr}^u$, following the off-the-shelf TTA methods. The optimal candidate model $\theta^{t*}$ is selected using $\arg\min_m \bar{\mathcal{L}}_{val}^t(\theta_m^*)$. After selecting the best model, we further refine it through supervised training, optimizing a cross-entropy loss $\mathcal{L}_{tr}^l$. The complete Human-in-the-Loop Test-Time Adaptation algorithm is detailed in Alg. 1.

## 4 Experiments

### 4.1 Experimental setting

**Datasets:** We select a total of five datasets for evaluation. The **CIFAR10-C** and **CIFAR100-C** (Hendrycks & Dietterich, 2018) are small-scale corruption datasets, with 15 different common corruptions, each containing 10,000 corrupt images with 10/100 categories. For our evaluation in large-scale datasets, we opt for **ImageNet-C** (Hendrycks & Dietterich, 2018), which also contains 15 different corruptions, each with 50,000 corrupt images in 1000 categories. Additionally, **ImageNet-D** (Rusak et al., 2022) is a style-transfer dataset, offering 6 domain shifts, each consisting of 10,000 images selected in 109 classes. Additionally, we evaluate our method on the **ModelNet40-C** dataset (Sun et al., 2022), which includes 3,180 3D point clouds affected by 15 common types of corruption.

**Implementation Details:** We evaluate TTA performance in the continual adaptation setting (Wang et al., 2022; Niu et al., 2022), where the target domain undergoes continuous changes. For CIFAR-10-C and

---

**Algorithm 1:** Human-in-the-Loop TTA

---

**Input** : Source Model $\theta_0$; Candidate Hyper-parameters $\Omega = \{\omega_m\}$; Testing Data Batches $\{\mathbb{B}_t\}$

**Output:** Predictions $\hat{\mathcal{Y}} = \{\hat{y}_i\}$

**for** $t = 1$ **to** $T$ **do**

  # Make Predictions:

  $\forall x_i \in \mathbb{B}_t,\ \hat{y}_i = h(x_i; \theta^{t-1*}),\ \hat{\mathcal{Y}} = \hat{\mathcal{Y}} \cup \hat{y}_i$

  # Oracle Annotation:

  Select labeled subset $\mathbb{B}_t^l$ by Eq. 8

  **for** $m = 1$ **to** $N_m$ **do**

    # Unsupervised Model Adaptation:

    $\theta_m^{t*} = \arg\min\limits_{\theta} \frac{1}{N_b} \sum\limits_{x_i \in \mathbb{B}_t^u} \mathcal{L}_{tr}^u(h(x_i; \theta); \omega_m)$

  # Update Moving Average Validation Loss:

  Update $\bar{\mathcal{L}}_{val}^t$ by Eq. 6.

  # Model Selection by Eq. 6:

  $\theta^{t*} = \arg\min\limits_{m} \bar{\mathcal{L}}_{val}^t(\theta_m^*)$

  # Supervised Model Adaptation:

  $\theta^{t*} = \arg\min_{\theta} \frac{1}{N_b} \sum\limits_{x_i, y_i \in \mathbb{B}_t^l} \mathcal{L}_{tr}^l(h(x_i; \theta), y_i; \omega_m)$

**return** *Predictions* $\hat{\mathcal{Y}}$;

---

CIFAR100-C datasets, we use a batch size of 200 and an annotation percentage of 3%, with pre-trained WideResNet-28 (Zagoruyko & Komodakis, 2016) and ResNeXt-29 (Xie et al., 2017) models, respectively. For ImageNet-C and ImageNet-D, we use a batch size of 64 for adaptation and an annotation percentage of 3.2% of the total testing samples, employing the ResNet-50 (He et al., 2016) pre-trained model.

We use the optimizer recommended in the original papers for each method's unsupervised training. We set the momentum $\beta$ for 0.5. For supervised training, we use the Adam optimizer with a learning rate of 1e-5. We consider seven distinct values for each hyper-parameter category regarding model selection, detailed in Tab. 8.

**Competing Methods:** We evaluate 6 state-of-the-art off-the-shelf test-time adaptation approaches under both traditional unsupervised TTA and Human-in-the-Loop TTA protocols, including TENT (Wang et al., 2020), PL (Lee et al., 2013), SHOT (Liang et al., 2020), EATA (Niu et al., 2022), SAR (Niu et al., 2023), RMT (Döbler et al., 2023). For each of the above TTA method under HILTTA protocol, we embed the TTA loss into the "Unsupervised Model Adaptation" step of the HILTTA algorithm in Alg. 1. We further benchmark against existing active TTA methods, including SimATTA (Gui et al., 2024) and VeSSAL (Saran et al., 2023). Specifically, we adapt VeSSAL for TTA by using the selected samples for supervised training. More details of the benchmarked methods are deferred to the Appendix.

## 4.2 Evaluations on HILTTA

We report the classification error rates for continual TTA in Tab. 2. For unsupervised TTA methods (**Unsup. TTA**), we report three results: worst(**Worst**), average (**Avg.**), and best (**Best**). The "Worst" and "Best" results are derived from fixed hyper-parameters that yield the highest and lowest overall error rate, respectively. The "Average" results are calculated by averaging the accuracy across models trained with all candidate hyper-parameters. For the competing active TTA methods (**Active TTA**), we maintain the same annotation rate as our method and use manually tuned hyper-parameters to ensure a fair comparison. These methods, lacking model selection capabilities, result in a single error rate per dataset. In contrast, our method (**Human-in-the-Loop TTA**) automatically selects hyper-parameters using our proposed model selection strategy, also resulting in a single error rate per dataset.

Table 2: Average classification error (lower is better) through ongoing adaptation to varying domains. "Worst", "Avg.", and "Best" indicate the worst, average, and best performance within the candidate hyper-parameter pool $\Omega$. The improvement of error rate for each off-the-shelf TTA method after combining with HIL is appended in the bracket as ($\delta$). * denotes methods under manually tuned hyper-parameters. $\diamondsuit$ denotes methods combined with our proposed HILTTA (with model selection). [†]We adapt VeSSAL for ATTA by combining TENT as an unsupervised training method.

| | TTA Method | ImageNet-C (%) ↓ | | | ImageNet-D (%) ↓ | | | CIFAR100-C (%) ↓ | | | CIFAR10-C (%) ↓ | | |
|---|---|---|---|---|---|---|---|---|---|---|---|---|---|
| | | Worst | Avg. | Best | Worst | Avg. | Best | Worst | Avg. | Best | Worst | Avg. | Best |
| **Unsup. TTA** | Source | | 82.02 | | | 58.98 | | | 46.44 | | | 43.51 | |
| | TENT (Wang et al., 2020) | 95.23 | 73.13 | 62.96 | 83.54 | 59.82 | 52.93 | 94.26 | 50.69 | 32.75 | 71.22 | 27.77 | 18.14 |
| | PL(Lee et al., 2013) | 89.78 | 81.48 | 67.75 | 63.06 | 56.59 | 52.45 | 37.31 | 34.67 | 32.76 | 19.95 | 19.11 | 18.54 |
| | SHOT(Liang et al., 2020) | 99.18 | 79.56 | 64.68 | 91.51 | 64.51 | 53.69 | 90.72 | 43.89 | 32.90 | 52.87 | 23.22 | 17.26 |
| | EATA(Niu et al., 2022) | 62.41 | 60.26 | 58.26 | 51.75 | 51.18 | 50.78 | 51.30 | 42.72 | 32.34 | 20.17 | 19.57 | 17.58 |
| | SAR(Niu et al., 2023) | 90.60 | 70.70 | 62.20 | 56.19 | 53.37 | 51.72 | 39.00 | 33.84 | 31.90 | 20.22 | 19.10 | 17.45 |
| | RMT(Döbler et al., 2023) | 98.77 | 72.28 | 60.45 | 88.80 | 58.92 | 50.14 | 38.35 | 32.18 | 30.16 | 29.36 | 19.75 | 16.66 |
| **Active TTA\*** | VeSSAL[†] (Saran et al., 2023) | | 62.78 | | | 49.77 | | | 45.97 | | | 17.47 | |
| | SimATTA (Gui et al., 2024) | | 64.09 | | | 49.26 | | | 31.75 | | | 17.66 | |
| **Human-in-the-Loop TTA (OURS)**$\diamondsuit$ | HIL + TENT | | 58.35 (+14.78) | | | 48.74 (+11.08) | | | 30.53 (+20.16) | | | 15.87 (+11.90) | |
| | HIL + PL | | 61.60 (+19.88) | | | 49.01 (+7.58) | | | 30.82 (+3.85) | | | 16.97 (+2.14) | |
| | HIL + SHOT | | 60.66 (+18.90) | | | 48.98 (+15.53) | | | 31.13 (+12.76) | | | **15.78** (+7.44) | |
| | HIL + EATA | | **55.13** (+5.13) | | | 47.51 (+3.67) | | | 30.93 (+11.79) | | | 16.40 (+3.17) | |
| | HIL + SAR | | 58.91 (+11.79) | | | 50.02 (+3.35) | | | 30.00 (+3.84) | | | 16.87 (+2.23) | |
| | HIL + RMT | | 58.26 (+14.02) | | | **46.28** (+12.64) | | | **29.41** (+2.77) | | | 16.45 (+3.30) | |

From the results, we make the following observations:

**i)** Existing unsupervised TTA methods, when applied without additional annotation, exhibit significant sensitivity to hyper-parameters. This sensitivity can lead to large variations in performance (e.g., TENT's best accuracy at 62.96% versus its worst at 95.23% on ImageNet-C). In the worst case, performance can even fall below the source model without any adaptation. In contrast, the "Best" performance across all methods are relatively close to each other. Nevertheless, selecting the best model without oracle knowledge is impractical in real-world scenarios. These findings highlight the persistent challenge of hyper-parameter sensitivity in TTA.

**ii)** Our proposed human-in-the-loop TTA consistently improves the performance of all off-the-shelf TTA methods against their unsupervised counterparts even with the best hyper-parameters. This demonstrates the robustness of our approach, mitigating the catastrophic failures that can arise from hyper-parameter sensitivity.

**iii)** Compared to existing active TTA approaches, even when manually tuning hyper-parameters to address the model selection issue, our proposed HILTTA consistently outperforms them across all four datasets. Notably, even a basic baseline like TENT achieves performance comparable to or better than existing active TTA methods when combined with our approach. These results highlight the effectiveness of integrating a small amount of labeled data through synergistic active learning and model selection, rather than relying solely on active learning.

## 4.3 Ablation & Additional Study

**Unveiling the Impact of Individual Components:** We conduct a comprehensive ablation analysis using TENT as the off-the-shelf TTA method in Tab. 3. Compared with not adapting to the testing data stream, fine-tuning with only the unsupervised training loss (Unsup. Train) could improve the performance, as expected. When human annotation is introduced, using the cross-entropy loss $\mathcal{H}_{ce}$ as validation objective (CE Valid.) does not guarantee a consistent improvement. As discussed in Zhao et al. (2023), this can be attributed to the challenges posed by the min-max equilibrium optimization problem across time, resulting in a notable performance decline. Incorporating additional anchor regularization for model selection (referred to as Anchor Reg.) consistently improves performance beyond the average error rate of the original TENT.

This underscores the necessity of stronger regularization for model selection to achieve overall good TTA performance. Additionally, integrating EMA smoothing (EMA Smooth.) of validation loss and supervised training with labeled data (referred to as Super. Train) both contribute positively to enhancing TTA performance.

Table 3: Ablation study of HILTTA with TENT (Wang et al., 2020) as off-the-shelf TTA method.

| Remark | Unsup. Train | CE Valid. | Anchor Reg. | EMA Smooth | Super. Train | ImageNet-C | ImageNet-D | CIFAR100-C | CIFAR10-C |
|---|---|---|---|---|---|---|---|---|---|
| | | | | | | Error Rate (%) ↓ | | | |
| Source | - | - | - | - | - | 82.02 | 58.98 | 46.44 | 43.51 |
| TENT | ✓ | - | - | - | - | 73.13 | 59.82 | 54.97 | 27.77 |
| w/ HIL | ✓ | ✓ | - | - | - | 90.52 | 62.66 | 67.82 | 21.36 |
| w/ HIL | ✓ | ✓ | ✓ | - | - | 65.95 | 52.87 | 47.55 | 17.79 |
| w/ HIL | ✓ | ✓ | ✓ | ✓ | - | 62.61 | 52.61 | 34.18 | 17.79 |
| w/ HIL | ✓ | ✓ | ✓ | ✓ | ✓ | **58.35** | **48.74** | **30.53** | **15.87** |

**Evaluation of Alternative Active Learning Strategies:** To optimize annotation resource allocation, we systematically evaluate various sample selection strategies within the HILTTA protocol, as illustrated in Fig.2. We explore multiple approaches, including random sampling (Random), entropy-based selection (Entropy(Shannon, 1948)), feature diversity sampling (CoreSet (Sener & Savarese, 2018)), stream-based active learning (VeSSAL (Saran et al., 2023)), the incremental clustering method introduced by Gui et al. (2024), and also our approach, K-Margin, uniquely combines uncertainty and feature diversity. Given that other active learning methods lack model selection capabilities, we employ cross-entropy loss $\mathcal{H}_{ce}$ as the validation objective. The results clearly demonstrate that K-Margin significantly outperforms other selection strategies in terms of average error rate across four datasets.

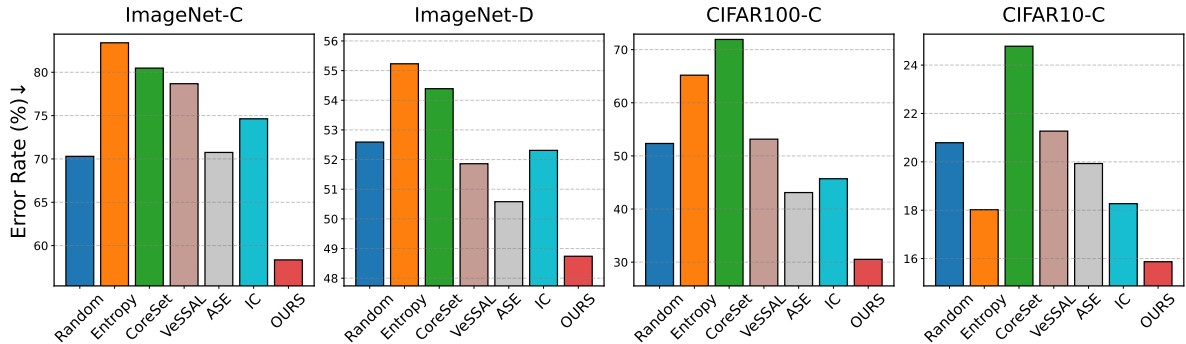

Figure 2: Comparison of different active learning strategies under HILTTA with TENT (Wang et al., 2020) as TTA unsupervised model adaptation strategy. Average classification error is reported, where a lower value indicates better performance.

**Evaluation of Alternative Model Selection Strategies:** We further compare our approach against several state-of-the-art model selection methods, including Entropy (Morerio et al., 2018), InfoMax (Musgrave et al., 2022), and MixVal (Hu et al., 2023), which are widely recognized in Unsupervised Domain Adaptation. Additionally, we consider the Oracle model selection strategy for online TTA, known as PitTTA (Zhao et al., 2023), which utilizes all labels within the streaming test dataset $\mathcal{D}_t$ to perform model selection based on accuracy comparison. The results presented in Tab.4 demonstrate that our proposed methods consistently surpass these alternative strategies, delivering comparable or superior performance to the best results obtained with fixed hyper-parameter settings. Notably, our HILTTA approach outperforms PitTTA(Zhao et al., 2023) while requiring significantly fewer annotations, making it a more practical and resource-efficient solution.

Table 4: Comparison of different model selection strategies under the HILTTA protocol with TENT (Wang et al., 2020) as unsupervised model adaptation strategy.

| Strategy | Error Rate (%) ↓ | | | | |
|---|---|---|---|---|---|
| | ImageNet-C | ImageNet-D | CIFAR100-C | CIFAR10-C | Avg. |
| Oracle Worst | 94.98 | 76.27 | 94.75 | 27.81 | 73.45 |
| Oracle Best | 58.49 | 48.55 | 30.11 | 15.75 | 38.23 |
| Entropy (Morerio et al., 2018) | 94.65 | 73.95 | 94.11 | 26.66 | 72.34 |
| InfoMax (Musgrave et al., 2022) | 94.56 | 77.00 | 93.14 | 28.81 | 73.38 |
| MixVal (Hu et al., 2023) | 92.98 | 68.20 | 92.23 | 19.64 | 68.26 |
| PitTTA (Zhao et al., 2023) | 60.75 | **47.95** | 47.99 | 16.23 | 43.23 |
| HILTTA (Ours) | **58.35** | 48.74 | **30.53** | **15.87** | **38.37** |

**Evaluations beyond 2D Image Recognition:** To further evaluate the effectiveness of our method beyond 2D image classification task, we conduct experiments on 3D point cloud classification using DGCNN (Wang et al., 2019) as the backbone, adapting it continuously across 15 domains within the ModelNet40-C (Sun et al., 2022) dataset. We maintained the same hyper-parameter candidate set as outlined in Tab. 8 and followed the experimental setup detailed in Tab. 2, with a batch size of 32 and an annotation percentage of 3.2%. The results, presented in Fig. 3, show that unsupervised TTA still consistently outperforms the baseline without any adaptation (Source). However, due to the sensitivity to hyper-parameters, all unsupervised TTA methods exhibit a substantial variation in performance with the "Best" and "Worst" hyper-parameters. All off-the-shelf TTA methods are consistently improved when integrated with our

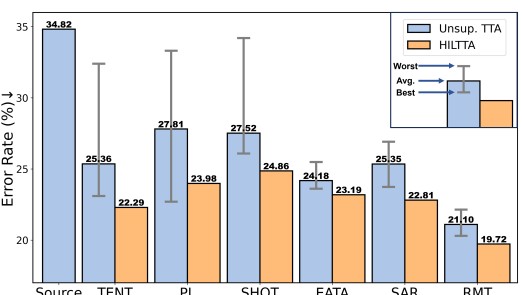

Figure 3: Performance on the 3D point cloud classification task in ModelNet40-C dataset. Average classification error is reported.

HILTTA method. Five out of the six methods witness a more significant boost in performance, surpassing their unsupervised counterpart with "Best" hyper-parameters.

**Computation Efficiency:** A major concern for test-time adaptation methods is the computation efficiency. We conduct empirical studies to measure the wall-clock time of both the inference and adaptation steps in HILTTA, utilizing a single RTX 3090 GPU, an Intel Xeon Gold 5320 CPU, and 30 GB of RAM. The results are presented in Tab. 5. In this experiment, we only perform human-in-the-loop annotation for every N batch on the testing data stream. The inference time required is fixed for all methods and the adaptation time varies. We demonstrate that the additional model selection procedure significantly decreases the error rate (27.77% → 15.87%) with 9 times the overall time required (1ms v.s. 9.1ms). More importantly, the overall time required can be further reduced by conducting sparser human annotation (larger N). With N=10, we witness a 10% absolute error rate decrease (27.77% → 17.13%) with only 2.8 times computation time. Considering that annotating a single image would cost 3000 ms to 5000 ms, the adaptation time is negligible.

Table 5: Computation efficiency under different human intervention frequencies.

| | w/o HIL | w/ HIL N=1 | w/HIL N=3 | w/HIL N=5 | w/HIL N=10 |
|---|---|---|---|---|---|
| Error Rate (%) | 27.77 | 15.87 | 15.98 | 16.43 | 17.13 |
| Inference Time (ms/sample) | 0.40 | 0.40 | 0.40 | 0.40 | 0.40 |
| Adaptation Time (ms/sample) | 0.60 | 8.70 | 3.50 | 2.60 | 1.40 |

**Detailed Analysis of Performance:** We provide a more detailed analysis against the selection of hyper-parameters. In particular, we investigate the performance of TENT with and without HIL and plot the results in Fig. 4. TENT w/o HIL and TENT w/o HIL (Average) refer to applying TENT without additional

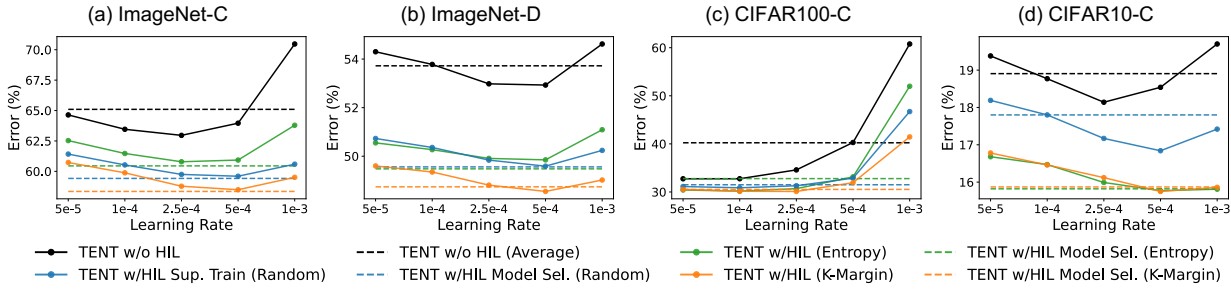

Figure 4: Performance of TENT (Wang et al., 2020) with and without HIL. Bold lines represent the performance with different hyper-parameter values, while dashed lines indicate the performance with model selection.

annotation and the average error rate respectively. TENT w/HIL Sup.Train (Random/Entropy/K-Margin) and TENT w/HIL Model Sel. (Random/Entropy/K-Margin) refer to augmenting TENT with supervised training on Random/Entropy/K-Margin labeled testing data and our proposed model selection respectively. The accuracy w/ HIL consistently surpasses all w/o HIL. Furthermore, random and entropy selection are consistently less effective than our proposed K-Margin selection for model selection purposes. Crucially, our final HILTTA approach consistently achieves significantly better performance than the worst hyper-parameter, and on ImageNet-C, we even surpass the model using the best-fixed hyper-parameter.

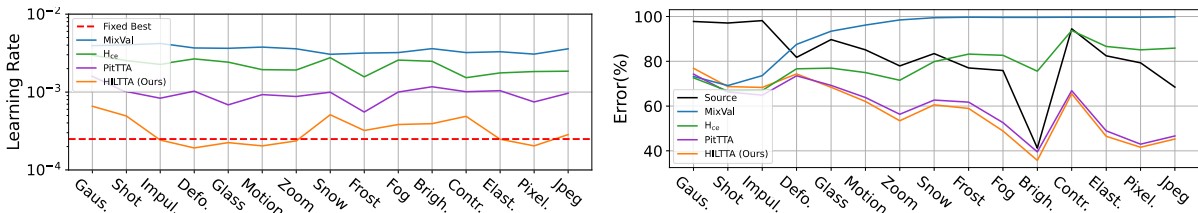

Figure 5: (Left) Average selected learning rate per corruption for TENT on ImageNet-C. (Right) Average error rate per corruption for TENT on ImageNet-C.

**Temporal Evolution Analysis of HILTTA:** We present an analysis of the evolution of selected hyper-parameters over time. As shown in Fig. 5, the hyper-parameters selected by our HILTTA model selection method always hover around the upper bound "Fixed Best" while competing methods deviate further away from the upper bound. The advantage is also reflected in the accumulated error plot. In particular, state-of-the-art model selection methods such as MixVal (Hu et al., 2023), are susceptible to catastrophic failure as they struggle to adapt to continuously changing domains.

Table 6: Select two hyper-parameters at the same time with PL on CIFAR10-C, error rate is reported.

| LR/Threshold | 0.1 | 0.2 | 0.4 | 0.6 | 0.8 | AVG. Error | HILTTA (OURS) |
|---|---|---|---|---|---|---|---|
| 1e-5 | 17.17 | 17.12 | 17.19 | 17.22 | 17.22 | | |
| 1e-4 | 17.08 | 17.10 | 17.01 | 17.13 | 17.04 | | |
| 1e-3 | 16.99 | 16.76 | 16.54 | 16.49 | 16.43 | 28.96 | **16.41** |
| 1e-2 | **15.89** | **15.56** | 20.29 | 23.69 | 23.56 | | |
| 1e-1 | 22.41 | 81.89 | 88.64 | 88.89 | 88.78 | | |

**Selecting Two hyper-parameters:** In certain cases, more than one hyper-parameter needs to be selected. We evaluated HILTTA combined with PL (Lee et al., 2013) on CIFAR10-C by selecting two hyper-parameters: the self-training threshold and the learning rate. As shown in Tab. 6, our HILTTA achieved the 3rd best result among 25 candidate hyper-parameter sets. These results suggest that our method is still effective with more than one hyper-parameters.

**Performance Under Different Labeling Budgets:** We investigate the effectiveness of HILTTA under different annotation budgets. Specifically, we vary the annotation ratio from 0% to 10%. As shown in Fig. 6, with the increase in annotation budget, we observe consistent improvement for all off-the-shelf TTA methods. Importantly, all methods can benefit significantly with only 1% labeled data. This suggests that limited labeled data could substantially ease the challenges of TTA.

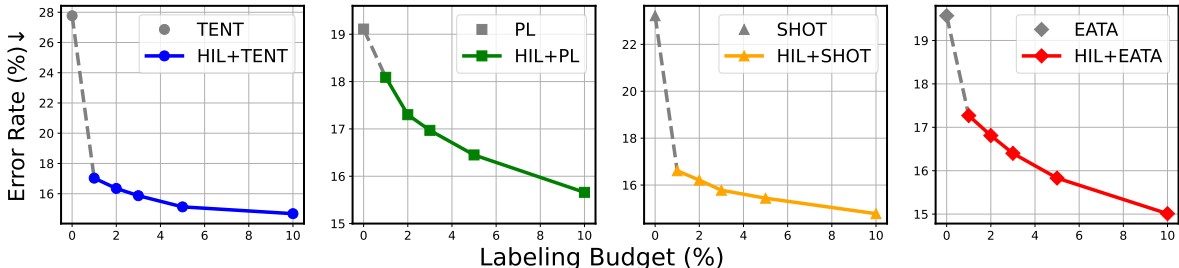

Figure 6: Performance of different TTA methods combined with our proposed HILTTA under different labeling budgets on CIFAR10-C, average error rates are reported (lower is better).

**Performance Under Adjustable Human Intervention Frequency:** We investigate the effectiveness of HILTTA under varying human intervention frequencies (N), as shown in Tab. 7. Instead of requesting human intervention for every batch of test streaming data, we reduce the frequency by only requesting intervention for every N-th batch. Consequently, the label rate is reduced to $1/N$. Model selection is also performed in every N batch, while for batches without human intervention, the previously selected hyperparameters are used for test-time adaptation. Remarkably, even when $N = 10$, corresponding to a mere 0.3% label rate, HILTTA combined with TENT significantly outperforms the original TENT without human intervention, achieving an error rate of 18.61% compared to 27.77%. This demonstrates that even with a very limited labeling budget and sparse human intervention, notable performance gains can be achieved. By employing an adjustable human intervention frequency, there is no need to annotate every batch, further enhancing HILTTA's practical impact.

Table 7: Performance of TENT (Wang et al., 2020) with HILTTA under different human intervention frequency (N) on CIFAR10-C.

| Human Intervention Frequency (N) | W/O HIL | 1 | 2 | 3 | 4 | 5 | 6 | 7 | 8 | 9 | 10 |
|---|---|---|---|---|---|---|---|---|---|---|---|
| Error Rate (%) | 27.77 | 15.87 | 16.43 | 17.15 | 17.53 | 17.94 | 17.99 | 18.00 | 18.05 | 18.39 | 18.61 |

## 5 Conclusion

In this study, we introduce a novel approach to human-in-the-loop test-time adaptation (HILTTA) by integrating active learning with model selection. Beyond utilizing active learning, we leverage the labeled data as a validation set, facilitating the smooth selection of hyper-parameters and enabling supervised training on a subset of the data. We also developed regularization techniques to enhance the validation loss and proposed a new sample selection strategy tailored for HILTTA. By integrating HILTTA with multiple existing TTA methods, we observed consistent improvements due to the synergistic effects of active learning and model selection, outperforming both the worst-case hyper-parameter choices and average performance, as well as existing active TTA methods. Our findings offer new insights into optimizing limited annotation budgets in TTA tasks.

**Acknowledgement**: This research is supported by the Agency for Science, Technology and Research (A*STAR) under its MTC Programmatic Funds (Grant No. M23L7b0021), the National Natural Science Foundation of China (NSFC) under Grant 62106078, and Sichuan Science and Technology Program (Project No. 2023NSFSC1421).

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

# A Experiment Details

## A.1 Detailed Settings for Datasets

Following prior works (Döbler et al., 2023; Wang et al., 2022; Niu et al., 2022), we perform continual test time adaptation in the following datasets, detailed in follows.

**CIFAR10-C** (Hendrycks & Dietterich, 2018) is a small-scale corruption dataset, with 15 different common corruptions, each containing 10,000 corrupt images of dimension (3, 32, 32) with 10 categories. We evaluate severity level 5 images, with "Gaussian → shot → impulse → defocus → glass → motion → zoom → snow → frost → fog → brightness → contrast → elastic → pixelate → jpeg" sequence.

**CIFAR100-C** (Hendrycks & Dietterich, 2018) is a small-scale corruption dataset, with 15 different common corruptions, each containing 10,000 corrupt images of dimension (3, 32, 32) with 100 categories. We evaluate severity level 5 images, with "Gaussian → shot → impulse → defocus → glass → motion → zoom → snow → frost → fog → brightness → contrast → elastic → pixelate → jpeg" sequence.

**ImageNet-C** (Hendrycks & Dietterich, 2018) is a large-scale corruption dataset, with 15 different common corruptions, each containing 50,000 corrupt images of dimension (3, 224, 224) with 1000 categories. We evaluate the first 5,000 samples with severity level 5 following Croce et al. (2021). The domain sequence is "Gaussian → shot → impulse → defocus → glass → motion → zoom → snow → frost → fog → brightness → contrast → elastic → pixelate → jpeg".

**ImageNet-D** (Rusak et al., 2022), as a large-scale style-transfer dataset, is built upon DomainNet (Peng et al., 2019), which encompasses 6 domain shifts (clipart, infograph, painting, quickdraw, real, and sketch). The dataset focuses on samples belonging to the 164 classes that overlap with ImageNet. Following Marsden et al. (2024b), we specifically select all classes with a one-to-one mapping from DomainNet to ImageNet, resulting in a total of 109 classes and excluding the quickdraw domain due to the challenge of attributing many examples to a specific class. After processing, it contains 5 domain shifts with each 10,000 images selected in 109 classes. The domain sequence is "clipart → infograph → painting → real → sketch".

**ModelNet40-C** (Sun et al., 2022), as a 3D point cloud corrupted dataset for benchmarking point cloud recognition performance under corruptions, is built upon ModelNet40 (Wu et al., 2015). We use the highest corrupted level 5 to evaluate the performance. The domain sequence is "background → cutout → density → density inc → distortion → distortion rbf → distortion rbf inv → gaussian → impulse → lidar → occlusion → rotation → shear → uniform → upsampling".

## A.2 Competing Methods

We mainly follow the official implementation of each method and use the same optimization strategy. But in certain cases, such as transferring from another protocol, we make minor changes to the configuration.

**Source** serves as the baseline for inference performance without any adaptation.

**TENT (Wang et al., 2020).** It focuses on updating batch normalization layers through entropy minimization. We follow the official implementation[1] of TENT to update BN parameters.

**PL (Lee et al., 2013).** We implement self-supervised training on unlabeled data updating all parameters by cross-entropy with an entropy threshold $E = \theta \times log(\#class)$. For CIFAR10-C and CIFAR100-C, we use an SGD optimizer with a 1e-3 learning rate, while an SGD optimizer with a 2.5e-5 learning rate for the rest.

**SHOT (Liang et al., 2020).** It freezes the linear classifier and trains the feature extractor by balancing prediction category distribution, coupled with pseudo-label-based self-training. We follow its official code[2] to update the feature extractor but remove the threshold. For CIFAR10-C and CIFAR100-C, we use an SGD optimizer with a 1e-3 learning rate, while an SGD optimizer with a 2.5e-5 learning rate.

---

[1] https://github.com/DequanWang/tent
[2] https://github.com/tim-learn/SHOT

**EATA (Niu et al., 2022).** It adapts batch normalization layers using entropy minimization but introduces an additional fisher regularization term to prevent drastic parameter changes. We follow the official implementation[3] of EATA to update BN parameters. We hold 2000 fisher training samples and keep the entropy threshold $E_0 = 0.4 \times log(\#class)$.

**SAR (Niu et al., 2023).** It updates batch normalization layers with a sharpness-aware optimization approach. We mainly follow its official code[4], but set $e_0 = 0$ for CIFAR10-C and CIFAR100-C, preventing failure cases of frequently resetting the model's weight.

**RMT (Döbler et al., 2023).** It adapts in a teacher-student mode, incorporating cross-entropy consistency loss and contrastive loss. We follow the official implementation[5].

**ASE** (Kossen et al., 2022). It uses surrogate estimators to sample in a uncertainty distribution to sample with low bias. We follow its official implementation[6].

**SIMATTA (Gui et al., 2024).** As an active test-time adaptation method, it selects labeled samples through incremental clustering and updates the model by minimizing cross-entropy. To reduce the computational and memory cost of incremental clustering, we keep the maximum length of anchors to 50 in all experiments. For CIFAR10-C and CIFAR100-C, we set the lower entropy bound $e_l$ to 0.005 and 0.01 for the higher entropy bound $e_u$. While for ImageNet-C and ImageNet-D, we set the lower entropy bound $e_l$ to 0.2 and 0.4 for the higher entropy bound $e_u$.

**VeSSAL (Saran et al., 2023).** As a stream-based active learning method, it uses volume sampling for streaming active learning. We also follow its official implementation[7], using feature embedding for stream-based data selection.

### A.3 Hyper-parameter Candidate set for Model Selection

Following Musgrave et al. (2022); Hu et al. (2023), in order to evaluate the performance of our method on different types of hyper-parameters, we construct candidate sets $\Omega$ with seven distinct values for each hyper-parameter category related to model selection. We assess different hyper-parameters for each TTA method: for PL and EATA, the self-training threshold; for TENT and SAR, the learning rate; and for SHOT and RMT, the loss coefficient factor. To ensure generality, we use the same $\Omega$ across diverse datasets, as summarized in Tab. 8.

Table 8: Overview of the TTA methods validated and their associated hyper-parameters

| TTA method | Hyper-parameter | Search Space |
|---|---|---|
| TENT (Wang et al., 2020) | Learning Rate $\eta$ | {5e-5, 1e-4, 2.5e-4, 5e-4, 1e-3, 2.5e-3, 5e-3} |
| PL (Lee et al., 2013) | Entropy Threshold $\tau$ | {0.05, 0.1, 0.2, 0.4, 0.6, 0.8, 1.0} |
| SHOT (Liang et al., 2020) | Loss Coefficient $\beta$ | {0.1, 0.25, 0.5, 1.0, 2.5, 5.0, 10.0} |
| EATA (Niu et al., 2022) | Learning Rate $\eta$ | {5e-5, 1e-4, 2.5e-4, 5e-4, 1e-3, 2.5e-3, 5e-3} |
| SAR (Niu et al., 2023) | Learning Rate $\eta$ | {5e-5, 1e-4, 2.5e-4, 5e-4, 1e-3, 2.5e-3, 5e-3} |
| RMT (Döbler et al., 2023) | Loss Coefficient $\lambda_{CL}$ | {0.1, 0.25, 0.5, 1.0, 2.5, 5.0, 10.0} |

---

[3]https://github.com/mr-eggplant/EATA
[4]https://github.com/mr-eggplant/SAR
[5]https://github.com/mariodoebler/test-time-adaptation
[6]https://github.com/jlko/active-surrogate-estimators
[7]https://github.com/asaran/VeSSAL

# B    Additional Experimental Analysis

## B.1    Ablation Study on the Impact of $\beta$

We evaluate the impact of the moving average momentum parameter, $\beta$, in Eq. 6 when applying TENT (Wang et al., 2020) combined with our proposed HILTTA. The experiments were conducted across four datasets: ImageNet-C, ImageNet-D, CIFAR100-C, and CIFAR10-C. We varied $\beta$ over a wide range, from 0.1 to 0.9, and observed in Fig. 7 that the performance remained robust throughout. This demonstrates that our proposed HILTTA method is not sensitive to the choice of $\beta$ values, maintaining stability across different settings.

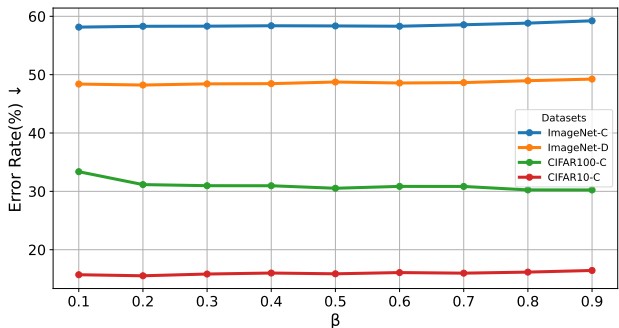

Figure 7: Performance of TENT (Wang et al., 2020) combined with our proposed HILTTA under different $\beta$ on 4 different datasets.

## B.2    Comparison to Bi-level Optimization

In some scenarios, the cost of collecting annotated data can outweigh the computational expense, making hyperparameter optimization via bi-level optimization (Liu et al., 2018) a viable approach. However, bi-level optimization has limitations, particularly its sensitivity to the meta-learning rate (the learning rate in the outer loop). We compare our method, HILTTA, with the approach proposed in (Liu et al., 2018), using Adam as the meta-optimizer. The Higher package (Grefenstette et al., 2019) is adopted for meta optimization. As shown in Fig 8, while the best performance achieved by bi-level optimization is comparable to ours (59.34% vs. 58.35% error rate), bi-level methods can be prone to collapse if the meta-learning rate is not carefully tuned. Additionally, bi-level optimization is unable to optimize hyperparameters that are non-differentiable.

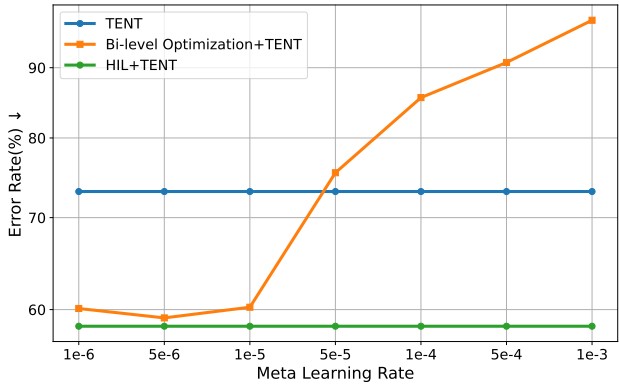

Figure 8: Performance of TENT (Wang et al., 2020) combined with our proposed HILTTA and bi-level optimization on the ImageNet-C dataset.

### B.3 Comparison to Semi-supervised Learning Methods

To evaluate the effectiveness of our proposed HILTTA, we compare it against semi-supervised methods, such as FixMatch (Sohn et al., 2020) and MeanTeacher (Tarvainen & Valpola, 2017), both using the same label rate of 3%. Additionally, we include ATTA methods like SimATTA (Gui et al., 2024) and VeSSAL (Saran et al., 2023), adapted to the ATTA setting by integrating them with TENT (Wang et al., 2020), also with label rate of 3%. As shown in Tab. 9, our HILTTA (HIL+TENT) surpasses all other TTA, semi-supervised learning, and ATTA approaches. Semi-supervised methods typically require longer training times and multiple epochs to achieve convergence, making them less effective in stream-based tasks. It is worth noting that while both semi-supervised and ATTA methods rely on manually tuned hyperparameters, HILTTA achieves superior performance with model selection.

Table 9: HILTTA Comparison with Semi-supervised Learning and Active TTA methods.

| Setting | TTA | Semi-supervised Learning | | ATTA | | HILTTA(Ours) |
|---|---|---|---|---|---|---|
| Method | TENT | FixMatch | MeanTeacher | SimATTA | VeSSAL | HIL + TENT |
| Error Rate(%) | 27.77 | 17.03 | 16.59 | 17.66 | 17.47 | **15.87** |

Although semi-supervised learning and ATTA methods can achieve performance comparable to HILTTA, they suffer from significant sensitivity to hyperparameters. As shown in Fig. 9, we compare the performance of TENT, HIL+TENT, and FixMatch on the CIFAR10-C dataset with learning rate as the variable hyperparameter. The results indicate that semi-supervised methods such as FixMatch are particularly sensitive to changes in learning rate, highlighting the challenges of tuning these methods effectively. Our proposed HILTTA is not only aimed at improving performance but, more importantly, at addressing hyperparameter sensitivity issues by synergizing active learning with model selection.

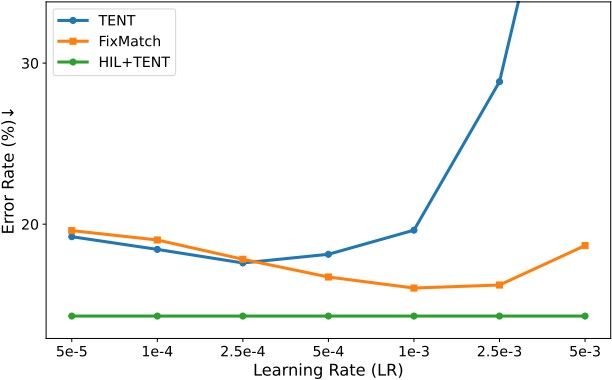

Figure 9: Performance of Fixmatch (Sohn et al., 2020) and TENT (Wang et al., 2020) combined with our proposed HILTTA on the CIFAR10-C dataset.

### B.4 Studies on the Role of Regularization in HILTTA

We evaluate the impact of the proposed anchor regularization and EMA smoothing techniques for model selection within the HILTTA framework. These methods are designed to prevent the model from overfitting to the local data distribution in the testing data stream. While removing these regularizations may improve adaptation to a single domain, it negatively impacts performance across future domains. As shown in Tab. 10, in the continual TTA setting, model selection without regularization results in significantly worse performance compared to when regularization is applied (17.25% vs. 15.87%). This indicates that regularization is crucial for maintaining robust performance in continual TTA. , model selection without regularization leads to significantly worse results (17.25% vs. 15.87%), highlighting the necessity of regularization for robust performance in continual TTA.

Table 10: Comparison of HILTTA with TENT on CIFAR10-C, showing error rates with and without regularization.

| Regularization | Time t ⟶ | | | | | | | | | | | | | | | Avg. |
| | gaussian | shot | impulse | defocus | glass | motion | zoom | snow | frost | fog | brightness | contrast | elastic | pixelate | jpeg | |
|---|---|---|---|---|---|---|---|---|---|---|---|---|---|---|---|---|
| Without | **25.04** | 20.23 | 28.57 | 12.22 | 29.67 | 13.56 | 11.45 | 15.57 | 14.59 | 13.64 | 8.26 | 11.29 | 20.44 | 14.86 | 19.37 | 17.25 |
| With | 25.93 | **20.13** | **27.49** | **11.59** | 27.69 | **12.48** | **9.54** | **14.04** | **13.01** | 12.06 | **6.59** | **8.52** | **18.21** | **12.46** | **18.26** | **15.87** |

## B.5 Comparing with Knowledge Distillation and Ensemble Learning

We carried out additional experiments by comparing with ensemble learning and multi-teacher knowledge distillation methods. Specifically, for ensemble learning, we average the posterior predicted by each candidate model and make a prediction with the average posterior. For multi-teacher knowledge distillation, we use the pseudo label predicted by the ensemble model for self-training. As observed from Tab. 11, both ensemble learning and multi-teacher knowledge distillation yields inferior results than HILTTA. This is most likely caused by incorporating poor candidate models (ensemble learning) and using poor pseudo labels for self-training, i.e. learning from noisy labeled data (multi-teacher knowledge distillation).

Table 11: Comparison with Ensemble Learning and Knowledge Distillation with TENT on CIFAR10-C.

| Method | Average Performance on all candidate model | Ensemble Learning | Muli-teacher Knowledge DIstillation | HILTTA (OURS) |
|---|---|---|---|---|
| Error | 27.77 | 19.48 | 26.42 | **15.87** |

## B.6 Comparing Adaptation Time & Annotation Time

We carried out empirical studies by measuring the wall-clock time lapses of the inference step (Inference Time), adaptation step (Adaptation Time), and human annotation (Annotation Time) per sample. In specific, the "Inference Time" refers to the time-lapse of pure forward pass, the "Adaption Time" refers to the time-lapse of backpropagation and weights update, and "Annotation Time" refers to the estimated time lapse required for human to annotate images at a 3% labeling rate. We evaluate under three categories of annotation cost, i.e. 2 sec/sample, 5 sec/sample, and 10 sec/sample. As seen from Tab. 12, despite the adaptation time being higher than the inference time (8.7 ms v.s. 0.4 ms), the annotation time is the most expensive step. These results suggest model selection and adaptation do not slow down the throughput under the HILTTA protocol.

Table 12: Comparing HILTTA and annotation time consuming with a 3% label rate on CIFAR10-C

| | Inference | Adaptation | Annotation (2 sec/sample) | Annotation (5 sec/sample) | Annotation (10 sec/sample) |
|---|---|---|---|---|---|
| Time (ms/sample) | 0.40 | 8.70 | 60.00 | 150.00 | 300.00 |

## B.7 Computation Complexity of Competing Active TTA Methods

We find that existing stream-based active learning (VeSSAL (Kossen et al., 2022)) and active TTA (SimATTA (Gui et al., 2024)) are inherently more computationally expensive than HILTTA due to the following reasons. First, VeSSAL adopts gradient embedding, i.e. $g(x_i) = \partial l(f(x_i; \theta), y_i)/\partial \theta_L$, where $\theta_L \in \mathbb{R}^{K \times D}$ refers to the classifier weights. This results in a very high dimensional gradient embedding $g(x_i) \in \mathbb{R}^{D \cdot K}$. The probability of labeling a sample $p_i$ involves calculating the inverse of covariance matrix, i.e. $p_i \propto g(x_i)^\top \hat{\Sigma}^{-1} g(x_i)$, $s.t. \hat{\Sigma} = \sum_{x_i \in \mathcal{B}} g(x_i)g(x_i)^\top \in \mathbb{R}^{DK \times DK}$. Hence, VeSSAL can hardly deal with classification tasks with a large number of classes, e.g. ImageNet, where $K$ is large and calculating a large matrix inverse is inherently time-consuming. Second, SimATTA employs a K-Means clustering algorithm for the selection of samples to annotate. Since K-Means is an iterative approach the computation cost of SimATTA is relatively high as well. In contrast, our proposed sample selection is built upon K-Center clustering which sequentially selects the sample with the highest distance to cluster centers (greedy algorithm), the computation cost is significantly lower. An

empirical study in Tab. 13 reveals that our method is about 3 times faster than VeSSAL and SimATTA in adaptation time.

Table 13: Error rate and adaptation wall time comparing to different methods on CIFAR10-C dataset

| Methods | Error | Adaptation Time (ms/sample) |
|---|---|---|
| VeSSAL | 24.41 | 26.90 |
| SimATTA | 17.66 | 22.50 |
| HILTTA (OURS) | **15.87** | **8.70** |

## B.8 Studies on Random Batchsizes

We recognize that in certain scenarios, test data may arrive inconsistently and intermittently. Results from Tab. 14 demonstrate that our proposed HILTTA framework is adaptable to such varying batchsize. At each iteration, we vary the batchsize randomly among the choices [50, 100, 150, 200]. This versatility leads to even greater improvements compared to fixed batch size. This enhanced performance stems from the fact that the optimal hyper-parameter values vary with different batch sizes, thus benefiting significantly from our adaptive model selection approach.

Table 14: Adaptation error with various batchsize with TENT on CIFAR10-C.

| Batchsize | w/o HIL | HILTTA |
|---|---|---|
| 200 | 27.77 | **15.87 (+11.90)** |
| Random choose from [50,100,150,200] | 35.89 | **16.39 (+19.50)** |

## B.9 Further Ablation Study with RMT

We have conducted wider experiments to analyze the ablation study with RMT (Döbler et al., 2023) as the off-the-shelf TTA method in Tab. 15 exhibits a consistent trend with the findings presented in the manuscript, providing additional support for our analysis and demonstrating the effectiveness of our proposed approach.

Table 15: Ablation study of HILTTA. Average classification error is reported for combing with RMT.

| Remark | Unsup. Train. | CE Valid. | Anchor regular. | EMA Smooth. | Super. Train. | Error (%) ↓ | | | |
|---|---|---|---|---|---|---|---|---|---|
| | | | | | | ImageNet-C | ImageNet-D | CIFAR100-C | CIFAR10-C |
| Source | - | - | - | - | - | 82.02 | 58.98 | 46.44 | 43.51 |
| RMT | ✓ | - | - | - | - | 72.28 | 58.92 | 32.18 | 19.75 |
| w/ HIL | ✓ | ✓ | - | - | - | 98.34 | 49.98 | 36.28 | 24.94 |
| w/ HIL | ✓ | ✓ | ✓ | - | - | 67.98 | 51.40 | 31.53 | 19.18 |
| w/ HIL | ✓ | ✓ | ✓ | ✓ | - | 61.33 | 50.23 | 30.75 | 17.33 |
| w/ HIL | ✓ | ✓ | ✓ | ✓ | ✓ | **58.26** | **46.28** | **29.41** | **16.45** |

