# OpenReview forum: "Exploring Human-in-the-Loop Test-Time Adaptation by Synergizing Active Learning and Model Selection"
_TMLR — Accepted by TMLR_

### Review · Reviewer_VRqh · 2024-09-14

**Summary Of Contributions:**

This paper proposes HILTTA, which combines Test-Time Adaptation with Human-in-the-Loop to stably select good hyperparameters and improve accuracy. The proposed method consistently outperforms various TTA-related approaches under ImageNet variant domain shifts.

**Audience:**

Yes

**Claims And Evidence:**

Yes

**Requested Changes:**

- It would be better to elaborate the appeal message of the proposed method beyond accuracy and/or propose some countermeasures to alleviate the burden of human intervention.

**Strengths And Weaknesses:**

## Strengths

- The proposed method is reasonable.
- The proposed method shows good performance under many situations.
- The experiments are comprehensive. Many ablation studies have been conducted.

## Weaknesses

- The proposed method requires human intervention. Popular unsupervised TTA methods are easier to use and more attractive to practitioners than the proposed method.
- It's not surprising that (semi) supervised methods outperform unsupervised methods. It would be better to compare the proposed method with more supervised methods (such as the ones in the online learning literature.)

---

> ### Author Response · Authors · 2024-10-28
> **Response to Reviewer VRqh**
>
> Dear Reviewer VRqh,
>
> We appreciate your insightful feedback and positive remarks on HILTTA's structure, comprehensive experiments, and performance. Below, we address the specific points you raised regarding human intervention and the method’s appeal beyond accuracy.
>
> ## Response to Weaknesses
>
> 1. **Human Intervention Requirement**
>    While HILTTA does require human intervention, it was designed specifically to address a key limitation in unsupervised TTA methods: sensitivity to hyperparameters, which typically require extensive manual tuning. Unlike unsupervised TTA, where performance can be inconsistent across different hyperparameter settings, HILTTA synergizes active learning with model selection to stabilize model performance without the need for constant tuning. In **Section 4.3**, we demonstrate that HILTTA remains effective even with minimal human intervention—e.g., only 0.3% of the dataset labeled and interventions at one in every 10 batches. This adaptability makes HILTTA a practical and feasible option for real-world applications where hyperparameter robustness is essential but resources for intervention are limited.
>
> 2. **Comparison with Supervised Methods**
>    In response to your suggestion, we have expanded comparisons to include several semi-supervised learning methods and active TTA methods in **Appendix B.2**. This comparison shows that while supervised methods may provide high performance, they often require prolonged training times to converge and manually hyperparameter tuning, limiting their practicality for TTA. In contrast, HILTTA’s integration of active learning and model selection reduces hyperparameter sensitivity, allowing it to perform stably in dynamic environments with minimal tuning.
>
> ## Response to Requested Changes
>
> 1. **Expanding Appeal Beyond Accuracy**
>    HILTTA was designed not only to improve performance but also to address the critical issue of hyperparameter sensitivity in TTA. Previous TTA and semi-supervised methods rely heavily on carefully tuned hyperparameters, with results often varying based on these settings. By combining active learning and model selection, HILTTA minimizes sensitivity to hyperparameters by model selection on the labeled subset, providing consistent performance without requiring detailed tuning. This robustness is essential for practical TTA scenarios, where tuning is difficult, and is detailed further in **Appendix B.2 and Section 4.3**.
>
> 2. **Minimizing Human Intervention**
>    HILTTA is structured to require sparse human input, making it suitable for real-world conditions with limited resources for intervention. In **Section 4.3**, our study on varying intervention frequencies demonstrates that HILTTA can maintain robust performance even with very infrequent interventions, such as labeling only 0.3% of the dataset. By balancing minimal human input with effective model selection, HILTTA offers a practical approach that retains stability and reliability with reduced intervention requirements.
>
> We hope these responses clarify the contributions and broader applicability of HILTTA and provide a clear understanding of its practical advantages.
>
> Kind regards,
> The Authors

---

> > ### Comment · Reviewer_VRqh · 2024-10-29
> >
> > Thank you for the response.
> >
> > The response and revision are clear and addressed my concern.

---

### Review · Reviewer_bwks · 2024-10-14

**Summary Of Contributions:**

The paper proposes Human-in-the-Loop Test-Time Adaptation (HILTTA) by integrating active learning and model selection to address sensitivity to hyper-parameters in existing test-time adaptation (TTA) methods. The framework involves selecting samples for human annotation and using labeled data for optimal hyper-parameter selection. Regularization techniques are introduced to prevent overfitting to local distributions during model selection, and a tailored sample selection strategy balances active learning and model selection objectives. Experiments on five TTA datasets demonstrate that HILTTA significantly improves performance compared to state-of-the-art TTA methods, mitigating the impact of poor hyper-parameter choices.

**Audience:**

Yes

**Broader Impact Concerns:**

Not observed.

**Claims And Evidence:**

Yes

**Requested Changes:**

1. Theoretical discussions are required.
2. A setting comparison table regarding the differences between settings Streaming active learning, TTA, ATTA, HILTTA, etc. is required.

**Strengths And Weaknesses:**

Strengths:

1. The paper is well-structured, with a clear flow from problem definition to proposed solution and experiments.
2. TTA is a critical area for real-world applications.
3. Addressing hyper-parameter sensitivity to enhance the robustness of TTA methods in dynamic environments is interesting.
4. The empirical results are comprehensive, demonstrating significant improvements over baselines across various corrupted datasets.

Weaknesses:

1. The theoretical aspects of the method were not discussed sufficiently.
2. The distinction of the HILTTA and ATTA is the introduction of model selection processes, which is clearly stated in the paper, but the names themselves are not distinguishable enough. Furthermore, as what I observed, settings are mixed together, e.g., VeSSAL is adapted to TTA and denoted as ATTA. That may confuse or potentially mislead readers.

---

> ### Author Response · Authors · 2024-10-28
> **Response to Review bwks**
>
> Dear Reviewer bwks,
>
> Thank you for your detailed review and positive feedback on the clarity, structure, and empirical rigor of our paper. We are grateful for your insightful comments regarding the theoretical aspects and naming clarity of HILTTA, which have helped us further refine and strengthen the manuscript.
>
> 1. **Theoretical Aspects of the Method**
>    We appreciate your recommendation to provide more theoretical insights into our proposed method. While our primary focus in this paper is on demonstrating HILTTA’s empirical effectiveness, we agree that additional theoretical discussion can provide greater depth. To address this, we have expanded **Section 3.4** of the revised manuscript with more discussions of principles guiding our sample selection and model adaptation strategy. Specifically, we provide insights into how the margin-confidence weighted K-Center clustering balances active learning with model selection, reducing sample redundancy and preventing biased sampling. We also address why our approach outperforms methods like entropy-based selection, as deep learning models often exhibit low entropy due to overconfidence, making the top-1 and top-2 probability gap a more reliable confidence measure. Additionally, entropy becomes less reliable in tasks with many classes, where random variations can distort its effectiveness. This discussion serves to clarify the theoretical motivations behind HILTTA’s design, and we believe it enhances the understanding of its robustness and effectiveness.
>
> 2. **Distinguishing HILTTA from ATTA and Naming Clarity**
>    We recognize your concern regarding the naming and differentiation between HILTTA, ATTA, and other adaptation methods, as well as potential confusion arising from terminology and setting adaptations (e.g., VeSSAL being adapted to ATTA). In response, we have taken the following steps to improve clarity:
>
>    - **Introduction of a Setting Comparison Table**: We have added a new table in the **Introduction (page 2)**, which details the differences in settings and methodologies between Streaming Active Learning, TTA, ATTA, and HILTTA. This table includes distinctions in the aspects of active learning and model selection. We believe this addition provides readers with a clear reference to understand the unique aspects of each setting.
>
>    - **Clarification on VeSSAL and ATTA Terminology**: To further avoid any potential confusion, we have added explicit notes clarifying that VeSSAL has been adapted to the ATTA setting for comparison purposes in **Table 2**.
>
> Thank you once again for your valuable feedback, which has helped us enhance both the theoretical grounding and clarity of our manuscript.
>
> Kind regards,
> The Authors

---

### Review · Reviewer_6jpG · 2024-10-15

**Summary Of Contributions:**

This paper proposes a novel human-in-the-loop test-time adaptation (HILTTA) approach. The proposed approach tackles one of the pitfalls of TTA methods, i.e., their sensitivity to the selection of hyperparameters, by integrating active learning into model selection. The proposed algorithm is simple: (1) collecting a small validation set (active learning), (2) running unsupervised model adaptation on a hyper-parameter set, (3) selecting a model, and finally (4) learning the selected model in a supervised manner. The technical contributions are (i) proposing this procedure, (ii) proposing a method for active learning, and (iii) proposing a metric for model selection. The paper also provides extensive evaluations on real-world datasets, which validate the effectiveness of the proposed algorithm.

**Audience:**

Yes

**Broader Impact Concerns:**

I don't have the broader impact concerns on this paper.

**Claims And Evidence:**

Yes

**Requested Changes:**

### Major  Comment
- **Validating of the robustness to the selection of $\beta$.** The proposed method has a hyperparameter, $\beta$, in the EMA smoothing (Eq. (6)). Table 2 reports the importance of the EMA smoothing on the CIFAR100-C dataset. Since the proposed method tackles the sensitivity issue to the selection of the model hyperparameters and the EMA smoothing can be important on some datasets, the authors should validate the robustness of the proposed method to the selection of $\beta$.

- **Clarifying the problem setting or adding experimental results for gradient-based methods solving Eq. (1).** In real-world applications, collecting labels even on a small validation set might be too *time-consuming*. It is not realistic to expect human-experts to start annotating a requested validation set immediately when we ask them to annotate a small validation set. So, the cost of collecting annotated data can dominate the computational time in the whole process. Then, it can be *feasible* to use gradient-based methods (e.g., Liu et. al, 2018) to solve the problem Eq. (1). I think that the authors should add experimental results for gradient-based methods solving Eq. (1) or clarify the problem setting and motivation in this view point.

**Strengths And Weaknesses:**

### Strength
- The proposed approach is novel and sounds reasonable.
- Extensive evaluations including ablation study are provided. They validate the effectiveness of the proposed algorithm.
- This paper is well-organized and explains the motivation and the idea clearly.

### Weakness
- Lack of theoretical guarantees/analyses on the proposed approach.
- Lack of some validations on the proposed approach (see requested changes).

---

> ### Author Response · Authors · 2024-10-28
> **Response to Reviewer 6jpG**
>
> Dear Reviewer 6jpG,
>
> Thank you for your insightful and constructive feedback on our manuscript. We appreciate your positive assessment of our contributions, as well as the valuable points you raised in the "Weakness" and "Requested Changes" sections. We address these points in detail below.
>
> ## Response to Weaknesses
>
> 1. **Lack of Theoretical Guarantees/Analyses**
> 	We appreciate your recommendation to provide more theoretical insights into our proposed method. While our primary focus in this paper is on demonstrating HILTTA’s empirical effectiveness, we agree that additional theoretical discussion can provide greater depth. To address this, we have expanded **Section 3.4** of the revised manuscript with more discussions of the principles guiding our sample selection and model adaptation strategy. Specifically, we provide insights into how the margin-confidence weighted K-Center clustering balances active learning with model selection, reducing sample redundancy and preventing biased sampling. We also address why our approach outperforms methods like entropy-based selection, as deep learning models often exhibit low entropy due to overconfidence, making the top-1 and top-2 probability gap a more reliable confidence measure. Additionally, entropy becomes less reliable in tasks with many classes, where random variations can distort its effectiveness. This discussion serves to clarify the theoretical motivations behind HILTTA’s design, and we believe it enhances the understanding of its robustness and effectiveness.
>
> 3. **Additional Validations for the Proposed Approach**
>    Please refer to the following:
>
> ## Response to Requested Changes
>
> 1. **Robustness to the Selection of β**
>    We appreciate your suggestion to validate the robustness of HILTTA to the choice of β in the EMA smoothing. As described in **Appendix B.1**, our additional experiments demonstrate that HILTTA consistently maintains performance across a wide range of β values (0.1 to 0.9) on multiple datasets, including CIFAR100-C, ImageNet-C, ImageNet-D, and CIFAR10-C. This result supports the robustness of HILTTA against variations in β, aligning with our objective of minimizing hyperparameter sensitivity.
>
> 2. **Clarification on the Problem Setting and Applicability of Gradient-Based Methods**
>    We thank you for pointing out the practical limitations involved in collecting labeled data for a small validation set in real-world scenarios, where annotation costs can exceed computational expenses. In response, we have included a comparison between HILTTA and bi-level optimization approaches in **Appendix B.2**, specifically referencing the approach of Liu et al. (2018). Bi-level optimization, as you noted, can be a practical alternative in certain contexts, as it optimizes hyperparameterson a held-out validation set.
>
>    To address this, we conducted a comparative experiment on ImageNet-C using the Higher package [1] to optimize hyper-parameters, with Adam as the meta-optimizer. Our results show that while the bi-level optimization method achieved a similar performance to HILTTA (59.34% vs. 58.35% error rate), it also highlighted notable limitations. Specifically, bi-level methods are sensitive to the meta-learning rate in the outer loop, and performance can collapse if the rate is not finely tuned. Moreover, bi-level optimization faces challenges in optimizing non-differentiable hyperparameters, thus limiting its flexibility. We hope this comparison clarifies our motivation for employing a human-in-the-loop approach with HILTTA, which maintains adaptability and minimizes hyperparameter sensitivity, even when human intervention is sparse.
>
> 	In **Section 4.3**, we also explore HILTTA’s effectiveness under varying levels of human intervention. The results demonstrate that HILTTA achieves robust performance improvements even with a very limited labeling budget (e.g., 0.3% of the dataset) and sparse human intervention frequencies (e.g., one in every 10 batches). These findings indicate that HILTTA’s reliance on human input can be adjusted based on practical constraints, reducing annotation frequency while maintaining effective adaptation. We believe this feature of HILTTA strengthens its applicability in real-world scenarios where reducing human intervention is essential.
>
> We hope these additions address your concerns thoroughly and demonstrate the robustness and practicality of our approach. Thank you again for your feedback, which has significantly contributed to improving our manuscript.
>
> Kind regards,
> The Authors
>
> [1] Grefenstette E, Amos B, Yarats D, et al. Generalized inner loop meta-learning. arXiv preprint arXiv:1910.01727, 2019.

---

> > ### Comment · Reviewer_6jpG · 2024-11-12
> >
> > Dear authors,
> >
> > Thank you for your detailed responses. My concerns were addressed in responses and revision.

---

### Author Response · Authors · 2024-10-28
**General Response to Reviewers' Comments on Manuscript Revision - Part 2**

5. **Comparison with Semi-Supervised Learning Methods - Appendix B.3**
   In Appendix B.3 of the revised manuscript, we included additional experiments comparing HILTTA with semi-supervised learning methods, such as FixMatch and MeanTeacher. These methods generally require substabtial longer training iterations to converge, making them less suitable for stream-based tasks. In contrast, HILTTA outperforms these methods and ATTA approaches within the same adaptation steps. Our findings also show that semi-supervised methods are highly sensitive to hyperparameter values, while HILTTA’s integration of active learning and model selection addresses these sensitivities, making it more robust.

6. **New Table on Methodological Differences - Introduction, Page 2**
   We have included a new table in the Introduction (page 2) that illustrates the differences between HILTTA, Active Test-Time Adaptation (ATTA), and Test-Time Adaptation (TTA). Additionally, we clarified in Table 2 that VeSSAL has been adapted to the ATTA setting, not the original one.

7. **Expanded Literature on Active Testing - Section 2.2**
   We have added references to related active testing literature in Section 2.2 to provide a comprehensive overview of the field and situate our work within the existing research landscape.

We hope these revisions address the reviewers' concerns and demonstrate the robustness and practicality of our approach. We look forward to your further feedback.

[1] Ash J T, Zhang C, Krishnamurthy A, et al. Deep batch active learning by diverse, uncertain gradient lower bounds. ICLR 2020.

[2] Grefenstette E, Amos B, Yarats D, et al. Generalized inner loop meta-learning. arXiv preprint arXiv:1910.01727, 2019.

---

### Author Response · Authors · 2024-10-28
**General Response to Reviewers' Comments on Manuscript Revision - Part 1**

We appreciate the constructive feedback provided by the reviewers, which has significantly helped us enhance the clarity and depth of our manuscript. Below, we outline the specific revisions made in response to each comment:

1. **Enhanced Theoretical Discussion on Sample Selection Strategy (K-Margin) - Section 3.4**
   We have added a more detailed theoretical discussion on our K-Margin sample selection strategy, emphasizing its unique advantage in achieving synergizing active learning and model selection highlighted in Sect 3.4 of the revised manuscript. Specifically, K-Margin prioritises samples with high uncertainty and selecting these sample helps model training, a.k.a. active learning. K-Margin also prevents selecting repetitive high uncertainty samples via the K-Center clustering on confidence weighted samples. ALthough similar objective was achieved by gradient embedding [1], we argue that gradient embedding is not suitable for large-scale classification tasks, e.g. 1000 classes for ImageNet. In contrast, our proposed K-Margin is not restricted by this limitation.
   We also discuss how our approach compares favorably to alternative selection methods, such as entropy-based approaches. Deep learning models often exhibit overconfidence, leading to generally low entropy. Therefore, the gap between the top-1 and top-2 class probabilities often serves as a more reliable indicator of model confidence than entropy alone. Furthermore, entropy can become less distinctive in classification tasks with many categories, as random variations in non-ground-truth classes can distort entropy calculations. Our empirical findings, detailed in Sect.4.3, further validate the advantages of the probability margin approach.

2. **Analysis of Moving Average Momentum Parameter, $\beta$ - Appendix B.1**
   We have added experiments analyzing the impact of the moving average momentum parameter, $\beta$, across four datasets: ImageNet-C, ImageNet-D, CIFAR100-C, and CIFAR10-C. By varying $\beta$ from 0.1 to 0.9, we observed that our proposed HILTTA method remains robust, showing consistent performance. This result indicates that HILTTA’s stability is not sensitive to $\beta$, supporting its adaptability across different settings. The evaluations are included in Appendix B.1 of the revised manuscript.

3. **Comparison with Bi-Level Optimization Approaches - Appendix B.2**
	We compared HILTTA with gradient based bi-level optimization approach on the ImageNet-C dataset with TENT as base TTA learner. The bi-level optimization is implemented via the Higher package [2]. The Adam optimizer was adopted as the meta optimizer. We observe the best result achieved by the bi-level optimizer is 58.35%, which is still worse than our HILTTA method. Importantly, the bi-level optimization is very sensitive to the choice of meta optimizer learning rate. Thus an important hyper-parameter, i.e. meta optimizer learning rate, is still introduced. Moreover, some hyper-parameters are non-trivial to be optimized by gradient based methods, e.g. confidence threshold $\tau$ for self-training. The evaluations and discussion are included in Appendix B.2 of the revised manuscript.

4. **Effectiveness of HILTTA with Varying Human Intervention Frequencies - Section 4.3**
   We explored the performance of HILTTA under various human intervention frequencies (N) in Section 4.3 of the revised manuscript. Results show that even with a very limited labeling budget (0.3%) and sparse human intervention (1/10), significant performance gains were achieved. This flexible intervention frequency eliminates the need to annotate every batch, underscoring HILTTA's practical impact by reducing annotation effort while maintaining performance.

---

### Decision · Action_Editor_ApwB · 2024-11-17

**Recommendation:** Accept as is

**Comment:**

The effectiveness of the proposed method has been studied through experiments on multiple datasets, and ablation studies show that all components of the proposed approach contribute effectively to its performance.
The authors appropriately addressed concerns raised by each reviewer, such as sensitivity to $\beta$ and the proportion of human intervention.
As a result, all reviewers agreed to accept the paper.

**Audience:**

Test-Time Adaptation is an important technique in practical machine learning and falls within the scope of TMLR.

**Claims And Evidence:**

This paper introduces Human-in-the-Loop Test-Time Adaptation (HILTTA) to address the sensitivity of hyperparameters in existing test-time adaptation (TTA) methods.
HILTTA combines active learning and model selection to achieve the following:

1. Sample Selection and Human Annotation: Selects samples from test data that require annotation and assigns labels through human input.
2. Hyperparameter Optimization: Uses the labeled data to identify the best hyperparameters.
3. Better Sample Selection Strategies: Designs strategies that balance the goals of active learning and model selection.


The effectiveness of the proposed method has been validated through experiments on multiple datasets.
Additionally, ablation studies confirm that all components introduced in the proposed approach contribute effectively to its performance.

---

> ### Author Response · Authors · 2024-12-19
>
> Dear Action Editor,
>
> We sincerely appreciate your efforts in managing the review process and are grateful for the recognition of our work. We have submitted the camera-ready version of our paper.
>
> Thank you once again for your support.
>
> Best regards,\
> The Authors